# Cancer Patients at Risk for Medication-Related Osteonecrosis of the Jaw. A Case and Control Study Analyzing Predictors of MRONJ Onset

**DOI:** 10.3390/jcm10204762

**Published:** 2021-10-17

**Authors:** Antonia Marcianò, Ylenia Ingrasciotta, Valentina Isgrò, Luca L’Abbate, Saveria Serena Foti, Antonio Picone, Matteo Peditto, Gian Marco Guzzo, Angela Alibrandi, Giacomo Oteri

**Affiliations:** 1Department of Clinical and Experimental Medicine, University of Messina, 98124 Messina, Italy; 2Department of Biomedical and Dental Sciences and Morpho-functional Imaging, University of Messina, 98124 Messina, Italy; ylenia.ingrasciotta@unime.it; 3Department of Diagnostics and Public Health, University of Verona, 37134 Verona, Italy; valentina.isgro@univr.it (V.I.); luca.labbate@unime.it (L.L.); 4Academic Spin-Off “INSPIRE—Innovative Solutions for Medical Prediction and Big Data Integration in Real World Setting”, Azienda Ospedaliera Universitaria “G. Martino”, 98124 Messina, Italy; saveriaserena.foti@unime.it; 5Department of Medical Oncology, Humanitas Istituto Clinico Catanese, 95045 Misterbianco, Italy; antonio.picone@ccocatania.it; 6Postgraduate School of Oral Surgery, Department of Biomedical, Dental Sciences and Morphofunctional Imaging, University of Messina, 98124 Messina, Italy; matteo.peditto@unime.it (M.P.); gianmarco.guzzo@libero.it (G.M.G.); giacomo.oteri@unime.it (G.O.); 7Unit of Statistical and Mathematical Sciences, Department of Economics, University of Messina, 98124 Messina, Italy; Angela.alibrandi@unime.it

**Keywords:** medication-related osteonecrosis of the jaws, risk factors, adverse drug reaction

## Abstract

The goal of this investigation was to identify potential risk factors to predict the onset of medication-related osteonecrosis of the jaw (MRONJ). Through the identification of the multiple variables positively associated to MRONJ, we aim to write a paradigm for integrated MRONJ risk assessment built on the combined analysis of systemic and local risk factors. The characteristics of a cohort of cancer patients treated with zoledronic acid and/or denosumab were investigated; beyond the set of proven risk factors a new potential one, the intake of new molecules for cancer therapy, was addressed. Registered data were included in univariate and multivariate logistic regression analysis in order to individuate significant independent predictors of MRONJ; a propensity score-matching method was performed adjusting by age and sex. Univariate logistic regression analysis showed a significant effect of the parameters number of doses of zoledronic acid and/or denosumab (OR = 1.03; 95% CI = 1.01–1.05; *p* = 0.008) and chemotherapy (OR = 0.35; 95% CI = 0.17–0.71; *p* = 0.008). The multiple logistic regression model showed that breast, multiple myeloma, and prostate cancer involved a significantly higher risk compared to lung cancer; a significant effect of the combined variables number of doses of zoledronic acid and/or denosumab (OR = 1.03; 95% CI = 1.01–1.06); *p*-value = 0.03) and exposure to novel molecule treatment (OR = 34.74; 95% CI = 1.39–868.11; *p*-value = 0.03) was observed. The results suggest that a risk assessment paradigm is needed for personalized prevention strategies in the light of patient-centered care.

## 1. Introduction

Medication-related osteonecrosis of the jaws (MRONJ) can be defined as “a drug-related adverse reaction, characterized by progressive bone destruction and necrosis of both the mandible and the maxilla, in patients taking aminobisphosphonates or other drugs (such as antiresorptive and antiangiogenic), with no other predisposing conditions” [1]. In the setting of cancer, the antiresorptive medications zoledronic acid and denosumab prescribed to prevent skeletal-related events associated with solid tumor-related bone metastases and with lytic lesions associated to multiple myeloma have been associated to MRONJ, in view of their inhibitory effects on osteoclastic bone resorption and remodeling. MRONJ has been also associated with some antiangiogenic medications such us bevacizumab because of its inhibition of the endothelial growth factor (VEGF) signaling cascade [2,3]. As mentioned, antiresorptives are primary agents in the current pharmacological treatment of a variety of other skeletal conditions such as malignancies metastatic to bone and cancer-induced low bone density. Current understanding of the mechanisms by which antiresorptives exert their effects on osteoclast-mediated bone resorption include osteoclast apoptosis and the discovery of the RANK/RANK-L/OPG pathway. Bisphosphonates bind to the bone mineral and determine osteoclast apoptosis, preventing the inhibitory effect of mature osteoclasts; in particular, aminobisphosphonates inhibit the mevalonate pathway enzyme farnesyl diphosphate (FPP) synthase in osteoclasts (Ocs), which leads to reduced production of specific intracellular proteins necessary for osteoclast function and survival, resulting in cell apoptosis. Non-aminobisphosphonates then become incorporated into the phosphate chain of ATP-containing molecules, rendering them non-hydrolyzable and cytotoxic to osteoclasts. Instead, denosumab precludes the binding of RANK-L to its receptor RANK on the surface of osteoclast OC precursors essential for the proliferation, differentiation, and survival of OC, since it is RANKL that initiates a complex signaling pathway which modulates intracellular calcium oscillation and turns on osteoclastogenesis [4,5,6].

MRONJ aside, antiresorptives are generally well tolerated, with the exception of gastroesophageal irritation—although their administration requires adequate calcium and vitamin D intake before and during therapy [7].

It has been reported that MRONJ incidence ranges from 1.2% to 9.9% in patients exposed to antiresorptive agents, reaching 15–20% in some case studies [8,9,10,11,12]. Although exposure to zoledronic acid or denosumab is the primary risk factor for MRONJ in cancer patients, additional factors have been associated with MRONJ—but for most of these, their contribution in the co-occurrence of this reaction remains unclear. Predisposing oral factors to the development of MRONJ are receiving attention, and oral management is recommended for prevention, since it has proved effective in reducing the incidence of MRONJ [13,14,15]—but above all, MRONJ incidence may be influenced by the malignancy type/severity, as well as by the contemporaneous intake of anti-cancer drugs [16,17,18,19,20]. Therefore, a multidisciplinary evaluation of patients before the initiation of bone-modifying agents is recommended [21,22,23]. This benefit-risk will vary from patient to patient, depending on the individual’s risk of developing skeletal-related events (e.g., disease extent, location, and activity) and the presence of risk factors for MRONJ [24]. Thus, a better understanding of MRONJ predictors is needed to guide stratification of risk [25]. The aim of this case–control study was to identify possible independent systemic risk factors for the development of MRONJ in the setting of cancer, evaluating the role of the underlying malignant disease and the concurrent administration of different anti-cancer medications. This information could be useful in order to perform individual risk assessments for patients about to initiate bone-targeting agents for cancer therapy, and to plan appropriate dental treatment protocols to prevent the development of MRONJ—prioritizing the patient’s quality of life and management of their skeletal malignant disease.

## 2. Materials and Methods

### 2.1. Study Design and Data Source

A case-control study was performed. Data on potential risk factors for MRONJ were collected using patient medical records from the Electronic Health Records (EHRs) of the Unit of Oral Surgery, School of Dentistry, University of Messina, which contain longitudinal MRONJ patient consultation data which were retrospectively analyzed and used as a source of data for MRONJ group. Controls were selected from the lists of the Cancer Centers in Eastern Sicily, including patients at risk for MRONJ identified through patient medical charts.

### 2.2. Study Population

Patients who had been diagnosed with metastatic cancer or multiple myeloma were included. Criteria for inclusion in the study were age ≥ 18 years, diagnosis of bone metastases from solid tumors or multiple myeloma and use of zoledronic acid and/or denosumab. Patients with previous radiation in the head and neck area were excluded from the study. Recruitment was a two-step process involving patients treated at the Osteonecrosis of the Jaw Treatment Center, School of Dentistry, of the University of Messina, and coming from the local Cancer Centers in Eastern Sicily during the years 2008–2018, respectively enrolled in the case and controls groups. Records of all cancer patients with intravenous bisphosphonates- or denosumab-related MRONJ reported in the Electronic Health Records (EHRs) of the Unit of Oral Surgery, School of Dentistry, University of Messina were collected retrospectively and used as a source of data for the MRONJ group. Controls were enrolled among patients identified from the local cancer centers receiving denosumab or zoledronic acid during the same period at the same cumulative dose who did not progress to MRONJ. At the time of enrollment, at least two pair-matched control patients were assigned to each patient case [26] through random sampling of the target population from the designated cancer centers. A propensity score matching 1:1 was performed to adjust for confounding variables (i.e., sex and age) [27,28]. Fully anonymized data were reported in a specific dataset respecting patient’s privacy.

### 2.3. Study Variables

Information on potential risk factors relevant for the purpose of the study were collected retrospectively. Demographic data (i.e., age expressed as mean and standard deviation, sex expressed as percentage) and the primary cancer type were collected. The total number of zoledronic acid or denosumab administered doses was recorded and evaluated. Concurrent cancer treatments were collected and categorized as the following: (a) chemotherapy, (b) hormonal therapy and (c) novel molecules (including target therapies, combination of chemotherapy and target therapy, immunotherapy and radiopharmaceuticals; See Appendix A, Table A1). Categorical variables, referring to treatments, are considered to be non-mutually exclusive. Moreover, a characterization of the MRONJ cases has been added to identify MRONJ features such as anatomic location of exposed necrotic bone areas, stage of the disease according to our currently adopted classifications from the American Association of Oral and Maxillofacial Surgeons (AAOMS) and the Italian Society of Oral Medicine and Oral Pathology (SIPMO) [2,29], and potential triggers (oral/dental findings) were obtained when available.

### 2.4. Statistical Analysis

Categorical variables are expressed as number and percentage, continuous data are summarized by mean and standard deviation. Comparisons between MRONJ and control groups were performed using Chi-Squared test (with reference to categorical data) and z-test (for proportions). To investigate associations between the explanatory variables (potential risk factors) and MRONJ, univariate logistic regression analysis was performed and applied to matched data by the propensity score method [19,20]. Then, a multiple logistic regression model was estimated in order to individuate significant independent predictors of MRONJ onset; the covariates inserted in the model were age, sex, cancer type, administration, chemotherapy, hormonal therapy, and novel molecules. Additionally, we included all interaction terms of the first order for each treatment. Estimated odds with *p* < 0.05 was considered significant. Statistical analysis was performed with R Studio (ver.1.3, RStudio, Boston, MA, USA).

## 3. Results

Overall, N = 75 patients affected by MRONJ were enrolled in the case group and N = 171 cancer patients were enrolled in the control group. Female prevalence was observed in both groups (N = 45 (60%) cases and N = 137 (80.1%) controls, respectively). The mean age was 70 years (SD 64–76) and 60 years (SD 51–70) for MRONJ and control patients, respectively. Primary cancer type was evaluated. Among the MRONJ patients, N = 14 (18.7%) had multiple myeloma, N = 38 (50.6%) had metastatic breast cancer, N = 20 (26.6%) had metastatic prostate cancer, and N = 3 (4%) patients had metastatic lung cancer. Patients in the control group had multiple myeloma (N = 11; 6.4%), metastatic breast cancer (N = 128; 74.8%), metastatic prostate cancer (N = 20; 11.6%), and metastatic lung cancer (N = 12; 7.0%). Concerning the studied bone metastasis treatment drugs, zoledronic acid was administered to N = 55 (73.3%) MRONJ patients and to N = 144 (84.2%) control patients. Denosumab was administered to N = 20 (11.7%) MRONJ patients vs. N = 27 (15.8%) in the control group. The mean number of doses administered until the moment of the diagnosis was 23.5 (15.7%) and 18.3 (12.6%) for MRONJ case and controls, respectively. Concerning cancer medications, chemotherapy was administered in the MRONJ group; the majority of patients had been treated with traditional chemotherapeutic treatment schemes (42 (56.0) patients in the MRONJ group and 144 (84.2%) in the control group), while alternative schemes were administered to 21 (28.0%) subjects in the MRONJ group and 70 (40.9%) in the control group, respectively. Thirty-eight (50.7%) patients in the MRONJ group and 116 (67.8%) patients in the control group received hormonal therapy. Two patients with multiple myeloma enrolled in the control group received zoledronic acid alone. Characteristics of cases and controls are reported in Table 1.

Characteristics of MRONJ lesions (anatomic location, clinical stage, presence of inflammation/infection and local risk factors) are reported in Table 2.

According to AAOMS classification of MRONJ, N = 30 (40%) patients had a stage I MRONJ, N = 32 (42.7%) had a stage II disease and N = 13 (17.3%) patients had stage III MRONJ. MRONJ lesions were reported to be mainly symptomatic, with 82.7% of patients showing clinical signs of inflammation/suppuration. According to the SIPMO staging system, N = 7 (5.2%) patients had a stage Ia MRONJ and 23 (17.2%) had a stage Ib MRONJ, N = 4 (3.0%) had a stage IIa, N = 28 (21.0%) had a stage IIb, N = 2 (1.5%) patients had stage IIIa, and N = 11 (8.2%) had a stage IIIb MRONJ. Regarding the anatomic location of MRONJ, a higher number of patients had lesions appearing in the mandible (N = 52; 69.3%), followed by the upper maxilla (N = 16; 21.3%), and jaws (N = 7; 9.3%). A local risk factor (oral/dental finding) potentially triggering MRONJ onset was registered in 21 of the MRONJ (28%) patients. The most frequent trigger was tooth extraction (21.3%), followed by soft tissue injuries due to dental prosthesis use (5.3%), and peri-implantitis (1.3%). In the remaining cases (N = 54; 72%) the potential trigger was not identified.

The estimates from univariate and multiple logistic regression model were reported in Table 3 and Table 4, respectively.

Using univariate logistic model, we found a significant effect of the number of administered doses of zoledronic acid/denosumab (OR = 1.03; 95% CI = 1.01; 1.05; *p* = 0.008) and chemotherapy (OR = 0.35; 95% CI = 0.17–0.71; *p* = 0.008) on MRONJ onset (Table 3).

Examining the results via a multiple logistic regression model, we found that all examined cancer types (breast cancer, multiple myeloma, and prostate cancer) involved a significantly higher risk of MRONJ onset compared to the lung cancer type. A significant association between MRONJ onset and the number of administered doses of zoledronic acid/denosumab (OR = 1.03; 95% CI = 1.01–1.06; *p*-value = 0.032) and exposure to novel molecules treatment (OR = 34.74; 95% CI = 1.39–868.11; *p*-value = 0.030) was also observed (Table 4).

## 4. Discussion

In cancer patients, MRONJ risk assessment is needed for personalized preventive strategies and to evaluate the indications and contraindications of conservative dental care and oral surgery in patients about to start taking bone-targeting agents and during the treatment course. This study was conducted in order to explore the MRONJ predictors among a cohort of cancer patients receiving zoledronic acid and/or denosumab, evaluating the association between clinical characteristics and MRONJ onset. Individual patient assessment is needed in order to guide risk stratification and plan preventive dental procedures, including tooth extraction when necessary, before treatment initiation with denosumab or zoledronic acid [30,31,32]. Using univariate and multiple logistic models we found a significant effect of the predictor administration (number of administered doses of zoledronic acid/denosumab), confirming previous results from the literature reporting that the increase in risk related to bisphosphonates and denosumab administration is dose dependent [33], and that MRONJ incidence increases with increased duration of exposure to antiresorptive agents, confirming the known dose-dependent fashion [34,35]. Our study showed that breast, multiple myeloma and prostate cancer patients have a significantly higher risk of developing MRONJ than patients affected by lung cancer (Table 4). The metastasis pattern of breast cancer varies with the subtype and has a predilection to metastasise to the bone; the same seems to happen for prostate cancer when it stops responding to deprivation therapy. These findings could help the oral surgeon in tailoring the most appropriate dental/oral prevention and follow-up for individual patients, starting from cancer diagnosis, and may also be useful for stratifying patients taking antiresorptive agents when, for example, the need for antiresorptives is associated with osteoporosis induced by previously administered hormonal therapies [36,37].

Cancer type seems to play an important role in the incidence of MRONJ. In the SWOG0702 trial, analysis by cancer type demonstrated a higher 3-year risk in multiple myeloma patients (4.3 versus 2.9% for prostate cancer, 2.7% for lung cancer, and 2.4% for breast cancer) [38]. In a report by Rugani et al., the weighted prevalence of medication-related osteonecrosis of the jaw was 2.09% in the breast cancer group, 3.8% in the prostate cancer group, and 5.16% for multiple myeloma patients [39]. Recently, an incidence of about 0.8% in breast cancer patients has been observed [40]. Walter et al. also reported a lower prevalence in breast cancer patients compared to prostate cancer and multiple myeloma patients [41]. It has been reported that patients with prostate cancer have a three-fold higher risk of denosumab-associated MRONJ compared to those with other cancer types [24,42,43,44]. Qi et al. reported that the prevalence of denosumab-related MRONJ in patients with prostate cancer was higher compared to that in patients with non-prostate cancers, related to the longer median follow-up period for prostate cancer compared with other tumor types—suggesting that the variability in the prevalence of ONJ in different cancer types may be due to this variation [45]. In a recent study by Ikesue et al. with 374 patients examining the patient characteristics between the denosumab and zoledronic acid groups, the distribution of cancer types was significantly different between groups (*p* < 0.001) [46]. The increased risk of MRONJ in these patients may be attributable to the dose and frequency of administration. Cancer patients receive multiple agents that interfere with bone metabolism and may, therefore, cause or support the development of osteonecrosis [39]. To aid in interpretation of this study, anti-cancer medications have been classified into three categories: hormonal therapy, chemotherapy and novel molecules. The novel molecules considered in the present study are mostly target therapies, for which cases of MRONJ in patients taking these drugs in association with zoledronic acid/denosumab osteonecrosis of the jaws have already been reported previously [47,48,49,50]. Regarding anti-cancer medications using a multiple logistic regression model, a significant association between MRONJ development and exposure to chemotherapy and novel molecules treatment was observed. A multiple logistic regression model showed that the risk of developing MRONJ was significantly higher in patients that received novel anti-cancer molecules and chemotherapeutic treatment vs. those treated with traditional chemotherapies alone (Table 4). In this study, hormonal therapy seems not to be an independent risk factor for the development of MRONJ. Indeed, among the secondary drugs that possibly contribute to MRONJ development, hormonal therapy has been indicated as a confounding factor. Neha et al. reported that the signal generated for aromatase inhibitor-associated osteonecrosis of the jaw in the Food and Drug Administration Adverse Event Reporting System database can be false positive, since upon removing the reports of concomitantly administered drugs (bisphosphonates and denosumab), signal strength for letrozole, anastrozole and exemestane drastically decrease [51,52]. Regarding the utilization of chemotherapy, MRONJ has already been associated with anticancer agents, including classic chemotherapy agents [3,53]. The review by Shim et al. summarizes fifty-four reported cases of osteonecrosis associated with chemotherapy in cancer patients [54], as the presence of an immunosuppressive status poses a high risk of developing infection, and chemotherapy has cytotoxic effects on bone metabolism and vascularization [30,55]. In patients with multiple myeloma, the use of thalidomide increased the risk of MRONJ by 2.4-fold (*p* = 0.043) [56]. For these patients with prior multiple chemotherapy, regimens should be monitored for early symptoms of MRONJ [54]. In addition to well-known medications, MRONJ may be a major adverse reaction to several new-generation anticancer drugs due to unknown mechanisms [57,58]. To date, several medications have been somehow implicated in MRONJ on the basis of experience gained through isolated data, case series reports, and literature reviews [16,17,18,59,60]. Recent reports have suggested a relatively high MRONJ risk in patients with a combined administration of bisphosphonates and targeted drugs [30,49]. It is known, for example, that the combination of anti-VEGFR and bone antiresorptive agents may increase the risk of MRONJ [60,61,62]. Similarly, it was well confirmed that mTOR inhibitors have a strong immunosuppressive effect, which can lead to delayed healing of oral soft tissues and persistent infections favoring the onset of the osteonecrotic process [63,64]. Taking into account these observations and their potential etiologies, the following hypothesis has been developed for MRONJ onset: (1) a direct local effect represented by the epithelial damage [65,66,67,68,69]; (2) an indirect systemic effect exerted by the immunosuppressive action of anti-cancer treatments [70,71], since the new concept of osteoimmunology [72] was recently added to the previous etiopathogenetic theories on MRONJ development [73]; and lastly, (3) the hypothesis that new anticancer drugs may play a role in osteoclast differentiation by additionally affecting the RANKL-mediated cell cycle arrest, as supported by recent in vitro and in vivo data [74,75]. With several therapeutic options to treat oncologic patients, we expect to see an increase in long-surviving patients in the metastatic bone phase receiving antiresorptive drugs [76,77] and, consequently, an increase in the number of MRONJ cases in the oncologic setting [78]. Starting from these assumptions, a significant portion of oncologic patients will need preventive dental care. Furthermore, dental interventions may also be required during the course of bone resorptive therapy [79]. Thus, for those who have had a concurrent administration of bisphosphonates and new anti-cancer molecules, dentists should be aware of a potentially increasing risk of severe MRONJ [60]. It is the authors’ opinion that risk assessment of cancer patients about to initiate bone targeting agents should be the expression of a strategic alliance between the oral surgeon and the oncologist, by introducing information such as overall survival and performance status in a combined evaluation. Moreover, while history of invasive dental procedures or local trauma may be present, some MRONJ cases occur spontaneously without any preceding factors. In other cases, a symptom without a clear odontogenic cause can represent a prodromal sign of MRONJ (i.e., odontalgia not explained by an odontogenic cause, as it happens when a sensory neuropathy in the distribution of the inferior alveolar nerve or mental nerve occurs) [80].

Early predictors of outcome could reflect long-term prognosis and support clinical decision making with the application of a combined paradigm that allows the tailoring of prevention and treatment strategies for each of our MRONJ patients.

Some open issues pertain to the duration of the antiresorptive therapy—it was shown that the risk for osteonecrosis of the jaw increases with duration of bisphosphonate therapy, even if it is unclear whether there is sufficient evidence to support de-escalation as a standard of care [35,81].

These questions will likely be addressed for both breast and prostate cancer by the REaCT-BTA (ClinicalTrials.gov identifier: NCT02721433) and REDUSE (ClinicaTrials.gov, accessed on 16 October 2021) identifier: NCT02051218) trials [82].

In general, in the absence of contraindications and relevant side-effects, patients should continue denosumab once every 4 weeks without any changes in the application regimen, while for patients with stable bone metastases on zoledronic acid, extending the administration frequency from once every 4 weeks to once every 12 weeks appears to be reasonable [83].

However, the ONJ rate was not reduced with this alternative regimen [84].

Other key questions are the differences in the patient’s prognosis using different medications. A recent systematic review and meta-analysis by Limones et al. found no statistically significant differences in the prognosis of MRONJ cases due to denosumab or zoledronic acid (*p* = 0.163). However, individual studies reported that MRONJ cases have a slightly higher tendency to resolve, ranging from 18 to 50%, compared to ZA, which ranges from 8% to 43%. This might be related to the reversible mechanism inherent to denosumab, which is not found in bisphosphonates in general [85].

Results deriving from this investigation could be an important contribution in the developmental literature on MRONJ, especially because individualized MRONJ prevention strategies can only come from a thorough understanding of risk factors. In this analysis, good data collection is combined with adequate statistical treatment, and appropriate inferential analysis was carried out in order to investigate predictors of MRONJ development. Furthermore, the present study, in addition to the existing ones, takes into consideration a potential risk factor—the intake of new molecules for cancer therapy, which showed significant results. This study also has some limitations—one above all is the sample size. We observed a statistical significance for exposure to new molecules; however, we need to be very cautious about concluding that these results outline a relationship with risk increase. A further limitation is related to patient enrollment, as the included population was not homogeneously paired. This paper uses a propensity score method to address the selection bias that potentially confounds the effects of the explanatory variables in observational studies. Current clinical practice prompted us to consider a well-structured sampling design with higher numbers for future studies on personalized risk assessments for MRONJ. The current standard in pharmacovigilance is bivariate association analysis, disregarding the probable co-occurrence of adversity [72,86]. Based on the above, we hypothesized a cumulative risk model for MRONJ prediction. Among the future developments, there is a further patient enrollment aiming to have a population of cases and controls matched by sex and age. The study sample should consist of at least 200 individuals diagnosed with MRONJ. Through identification of a set of proven risk factors, we will aim to thereby combine the multiple variables positively associated with MRONJ into a single index to predict outcomes, and to be used to schedule individualized prevention strategies for patients at risk.

## 5. Conclusions

The pathogenesis of MRONJ is likely to be multifactorial and can involve a synergistic effect between exposure to bisphosphonates or denosumab, other anti-cancer agents, and local factors. Exposure to denosumab or bisphosphonates is the primary risk factor for MRONJ; nevertheless, besides the triggering events (dental extraction, periodontal infection, ill-fitting prostheses), there might be some other systemic determinant factors. Association of chemotherapy and/or new anti-cancer molecules, in sequence or as single therapies, may contribute to MRONJ development. Although no definitive conclusions have been reached regarding the influence of anti-cancer drugs on MRONJ development, these may represent an additional risk factor for the occurrence of MRONJ.

## Figures and Tables

**Table 1 jcm-10-04762-t001:** Characterization of cases and controls before and after adjustment.

	MRONJ Group	Controls Group	*p*-Value	MRONJ Group	Controls Group	*p*-Value
	N = 75 (%)	N = 171 (%)		N = 75	N = 75	
				(After Adjustment)	(After Adjustment)	
Age-Mean (SD)	68.3 (9.7)	60.5 (12.7)	< 0.001	68.3 (9.7)	68.8 (10.3)	0.757
Sex						
Male	30 (40.0)	34 (19.9)	0.002	30 (40.0)	22 (29.3)	0.230
Female	45 (60.0)	137 (80.1)		45 (60.0)	53 (70.7)	
Primary cancer type						
Multiple myeloma	14 (18.7)	11 (6.4)	0.007	14 (18.7)	9 (12.0)	0.365
Metastatic breast cancer	38 (50.7)	128 (74.9)	<0.001	38 (50.7)	46 (61.3)	0.250
Metastatic prostate cancer	20 (26.7)	20 (11.7)	0.006	20 (26.7)	13 (17.3)	0.237
Metastatic lung cancer	3 (4.0)	12 (7.0)	0.535	3 (4.0)	7 (9.3)	0.326
Bone metastasis treatment drugs						
Denosumab	20 (26.7)	27 (15.8)	0.069	20 (26.7)	11 (14.7)	0.107
Zoledronic acid	55 (73.3)	144 (84.2)	0.069	55 (73.3)	64 (85.3)	0.107
Mean number of doses administered (SD)	23.5 (15.7)	18.3 (12.6)	0.006	23.5 (15.7)	16.91 (12.6)	0.005
Cancer medications						
Chemotherapy	42 (56.0)	144 (84.2)	<0.001	42 (56.0)	60 (80.0)	0.003
Hormonal therapy	38 (50.7)	116 (67.8)	0.016	38 (50.7)	47 (62.7)	0.187
Novel molecules	21 (28.0)	70 (40.9)	0.073	21 (28.0)	23 (30.7)	0.858

Legend: MRONJ = medication-related osteonecrosis of the jaws; SD = standard deviation.

**Table 2 jcm-10-04762-t002:** Clinical characteristics of lesions in the MRONJ group.

MRONJ Staging (AAOMS Classification)	MRONJ Group N = 75 (%)
0	3 (2.25)
I	11 (8.25)
II	48 (36)
III	13 (9.75)
Presence of clinical signs of inflammation/suppuration at diagnosis stage	62 (82.7)
MRONJ staging (SIPMO classification)	
Ia	7 (5.25)
Ib	23 (17.25)
IIa	4 (3)
IIb	28 (21)
IIIa	2 (1.5)
IIIb	11 (8.25)
Anatomic location	
Mandible	52 (69.3)
Upper maxilla	16 (21.3)
Both jaws	7 (9.3)
Trigger	
Tooth extraction	16 (21.3)
Peri-implantitis	1 (1.3)
Soft tissue injuries due dental prosthesis	4 (5.3)
Unidentified trigger	54 (72)

Legend: MRONJ = medication-related osteonecrosis of the jaws; AAOMS = American Association of Oral and Maxillofacial Surgeons; SIPMO: Italian Society of Oral Medicine and Oral Pathology.

**Table 3 jcm-10-04762-t003:** Logistic regression model to identify potential predictors of risk of MRONJ in cancer patients.

Covariate	OR (CI 95%)	*p*-Value
Age	0.99 (0.96–1.03)	0.755
Sex (male)	1.60 (0.81–3.16)	0.171
Lung cancer	0.59 (0.16–2.17)	0.430
Breast cancer	0.65 (0.34–1.24)	0.189
Multiple myeloma	1.68 (0.68–4.16)	0.261
Prostate cancer	1.73 (0.79–3.80)	0.170
Number of administered doses of zoledronic acid/denosumab	1.03 (1.01–1.05)	**0.008**
Chemotherapy	0.35 (0.17–0.71)	**0.003**
Novel molecules	1.07 (0.52–2.19)	0.855
Hormonal therapy	0.65 (0.34–1.24)	0.189

Legend: OR = Odds Ratio; CI = Confidence Interval. Statistically significant *p*-values are reported in bold.

**Table 4 jcm-10-04762-t004:** Multiple logistic regression model to identify independent predictors of risk of MRONJ in cancer patients.

Covariate	OR (CI 95%)	*p*-Value
Age	0.97 (0.93–1.02)	0.234
Sex (male)	2.05 (0.50–8.42)	0.320
Cancer type (ref = lung cancer)		
Breast cancer	17.56 (1.43–215.05)	**0.025**
Multiple myeloma	16.28(1.49–181.98)	**0.022**
Prostate cancer	17.99 (1.60–204.68)	**0.019**
Number of administered doses of zoledronic acid/denosumab	1.03 (1.01–1.06)	**0.032**
Chemotherapy	1.06 (0.20–5.54)	0.944
Hormonal therapy	1.35 (0.22–8.18)	0.740
Novel molecules	34.74 (1.39–868.11)	**0.030**
Chemotherapy–hormonal therapy	0.24 (0.03–1.71)	0.156
Chemotherapy–novel molecules	0.03 (0.01–1.08)	0.055
Hormonal therapy–novel molecules	0.64 (0.09–4.65)	0.662

Legend: OR = Odds Ratio; CI = Confidence Interval. Statistically significant *p*-values are reported in bold.

## Data Availability

Anonymized data of all patients are archived in a database specifically created with the purpose of conducting this research. In order to build this database, the medical records of all eligible patients have been extracted from the electronic health records of the involved institutes.

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
