# Peer review of "Cancer Patients at Risk for Medication-Related Osteonecrosis of the Jaw. A Case and Control Study Analyzing Predictors of MRONJ Onset"

_jcm, 2021, doi:10.3390/jcm10204762_

Round 1
Reviewer 1 Report
The manuscript submitted to JCM entitled “Cancer patients at risk for medication related osteonecrosis of the jaws. A case and control study analyzing predictors of MRONJ onset” is an original article which aim to analyze the potential risk factors of medication-related osteonecrosis of the jaws (MRONJ) in a cohort of cancer patients treated with zoledronic acid and/or denosumab.
On my opinion the article is interesting, well written, with good English.
However, I highlighted some issues.
- English language: Minor spell check is required.
- Abstract: Please structure the abstract to attract the reader's attention and adapt it accordingly.
- Introduction: This section has been properly prepared.
- Material and methods: This section has been properly prepared.
- Results: This section has been properly prepared.
- Discussion: Please discuss the onset of numb chin syndrome (or mental neuropathy) as predictive factor of MRONJ onset [https://doi.org/1016/j.jormas.2018.04.006].
- Conclusions: This section has been properly prepared.
After making the indicated changes, the article will be suitable for publication.
Thanks for the opportunity to review this manuscript.
Author Response
We sincerely thank the reviewers for their thoughtful advices since we believe the suggested changes are an important improvement for our work and were highlighted in the text in green color.
Abstract has been modified as following in order to be more attractive for the reader:
The goal of this investigation was to identify potential risk factors to predict the onset of medication-related osteonecrosis of the jaw (MRONJ). Through the identification of the multiple variables positively associated to MRONJ we aim at the writing of a paradigm for integrated MRONJ risk assessment built on the combined analysis of systemic and local risk factors. The characteristics of a cohort of cancer patients treated with zoledronic acid and/or denosumab were investigated, beyond the set of proven risk factor a new potential ones, the intake of new molecules for cancer therapy, was addressed. Registered data were included in univariate and multivariate logistic regression analysis in order to individuate significant independent predictors of MRONJ, propensity score matching method was performed adjusting by age and sex. Univariate logistic regression analysis showed a significant effect of the parameters number of doses of zoledronic acid and/or denosumab (OR= 1.03; 95% CI= 1.01-1.05; p=0.008) and chemotherapy (OR= 0.35; 95% CI= 0.17-0.71; p=0.008). The multiple logistic regression model showed that breast, multiple myeloma, and prostate involved a significantly higher risk compared to lung cancer type; a significant effect of the combined variables number of doses of zoledronic acid and/or denosumab (OR=1.03; 95% CI=1.01-1.06); p-value=0.03) and exposure to novel molecules treatment (OR=34.74; 95% CI =1.39- 868.11; p-value=0.03) was observed. The results suggest that a risk assessment paradigm is needed for personalized prevention strategies in the light of patient centered care.
Furthermore the onset of numb chin syndrome has been discussed as suggested.
We have welcomed in a special way the reviewer's suggestion to discuss the onset of numb chin syndrome as an unusual clinical presentation of MRONJ.
We added the following:
While history of invasive dental procedures or local trauma may be present, some MRONJ cases occur spontaneously without any preceding factors. In other cases a symptom without a clear odontogenic cause can represent a prodromal sign of MRONJ. The odontalgia not explained by an odontogenic cause as it happens when a sensory neuropathy on the distribution of the inferior alveolar nerve or mental nerve occurs.
With proper citation:
Fortunato L, Amato M, Simeone M, Bennardo F, Barone S, Giudice A. Numb chin syndrome: A reflection of malignancy or a harbinger of MRONJ? A multicenter experience. J Stomatol Oral Maxillofac Surg. 2018 Nov;119(5):389-394. doi: 10.1016/j.jormas.2018.04.006. Epub 2018 Apr 20. PMID: 29680775.
We hope this meet the reviewer's expectations.
Reviewer 2 Report
The skeleton is the most common site to be affected by metastatic cancers, so. myeloma, breast, and prostate cancers harbor the highest prevalence of bone metastases. Administration of bisphosphonates are a good way to treat bone cancer since they inhibit normal and pathological osteoclast-mediated bone resorption by multiple mechanisms. The overall effect of bisphosphonate treatment is the collective reduction of excessive bone turnover, resulting in preservation of structure and mineralization of the bone
The bisphosphonates zoledronic acid and denosumab are two antiresorptive drugs which are in use for treating osteoporosis. They have different mechanisms of action, but both have been shown to delay the onset of skeletal-related events in patients with advanced cancer
In 2003, oral surgeons noticed that some patients receiving bisphosphonates were developing osteonecrosis of the yaw mainly after tooth extraction. Here the overall rate of osteonecrosis of the jaw was approximately 3% at 3 years but varied somewhat by cancer type.
Osteonecrosis was seen after administration of zoledronate and though it was hoped that denosumab which was later on the market would not be associated with osteonecrosis of the jaw. But unfortunately, studies showed that the incidence of DRONJ in cancer patients under treatment with denosumab ranged from 0.5 to 2.1% after 1 year, 1.1 to 3.0% after 2 years, and 1.3 to 3.2% after 3 years of exposure.
To the article:
The aim of the authors was to analyze the potential risk factors of medication-related Osteonecrosis of the jaws (MRONJ) of cancer patients treated with zoledronic acid and/or denosumab.
A case-control study was conducted during the years 2008-2018.
Introduction:
The authors give a short overview about medication with bisphosphonates.
I miss information on how bisphosphonates in general react in bone/osteoblasts/osteoclasts. Please give a short overview.
Why react different cancer types in a different way, please give a short overview; or is this still unknown?
What about smoking, do you have information? Do you know if smoking enhances the risk of chemo osteonecrosis? Do you have any information about smoking behavior of your collective?
Summarize the general risk factors in using bisphosphonates.
Materials and Methods:
Is described in a sufficient way.
Results and discussion:
Zoledronate was given to 55 patients and denosumab to 20 patients.
The results are presented in several tables.
As expected, and still known the most frequent trigger for developing osteonecrosis of the yaw was tooth extraction (21.3%), followed by soft tissue injuries due to dental prosthesis use (5.3%) and peri-implantitis (1.3%).
These findings are not quite new.
The study showed that breast cancer, multiple myeloma and prostate cancer patients had a significantly higher risk of developing MRONJ than patients affected by lung cancer.
Can you explain this by different pathological metastatic behavior of these cancer types?
Statistical methods are used in a sufficient way.
You say that "It has been reported that patients with prostate cancer have a three-fold higher risk of denosumab- associated MRONJ as compared to those with other cancer types [." Do you have any explanations? Please discuss this.
Some key questions that remain are:
How often, and how long, should patients with bone metastases be given zoledronic acid or denosumab? Do you have any information?
Do you know, if there were main differences in the patient’s prognosis using different medication?
Author Response
We sincerely thank the reviewers for their thoughtful advices since we believe the suggested changes are an important improvement for our work and were highlighted in the text.
In particular we thank the reviewer for the suggestion to add a short overview on how bone antiresorptives agents.
We added the following paragraph in the Introduction section: “As mentioned, antiresorptives are primary agents in the current pharmacological treatment of a variety of other skeletal conditions such as malignancies metastatic to bone and cancer induced low bone density. Current understanding of the mechanisms by which antiresorptives exert their effects on osteoclast-mediated bone resorption include osteoclast apoptosis and the discovery of the RANK/RANK-L/OPG pathway. Bisphosphonates binds to the bone mineral and determine osteoclast apoptosis preventing the inhibitory effect of mature osteoclasts; in particular aminobisphosphonates inhibit the mevalonate pathway enzyme farnesyl diphosphate (FPP) synthase in osteoclasts (Ocs) which leads to reduced production of specific intracellular proteins necessary for osteoclast function and survival resulting in cells apoptosis; non-aminobisphosphonates become incorporated into the phosphate chain of ATP-containing molecules, rendering them non-hydrolyzable and cytotoxic to osteoclasts. Instead denosumab precludes the binding of RANK-L to its receptor RANK on the surface of osteoclasts OC precursors essential for the proliferation, differentiation, and survival of OC since it is RANKL to initiates a complex signaling pathway which modulate intracellular calcium oscillation and turns on osteoclastogenesis”.
Wat WZM. Current Controversies on the Pathogenesis of Medication-Related Osteonecrosis of the Jaw. Dent J (Basel). 2016;4(4):38. Published 2016 Oct 28.
Fisher JE, Rodan GA, Reszka AA. In vivo effects of bisphosphonates on the osteoclast mevalonate pathway. Endocrinology. 2000 Dec;141(12):4793-6.
Chiu YG, Ritchlin CT. Denosumab: targeting the RANKL pathway to treat rheumatoid arthritis. Expert Opin Biol Ther. 2017;17(1):119-128.
In reply to the reviewer’s comment: on tobacco use this has been inconsistently reported as a risk factor for MRONJ thus we didn’t record this information when reporting our cohort patients characterstics. Indeed the question is relevant, in a first period on MRONJ pathophysiology understanding efforts this correlation was investigated therefore data on smoking habit are available for a limited number of MRONJ patients and in lung cancer patients at risk for MRONJ. On the other hand considering the smoking role in periodontitis progression it could indirectly enhance MRONJ onset risk through the periodontal disease trigger.
Johannsen A, Susin C, Gustafsson A. Smoking and inflammation: evidence for a synergistic role in chronic disease. Periodontol 2000. 2014 Feb;64(1):111–126.
The general risk factors in using bisphosphonates has been summarised as suggested adding the following sentence:
MRONJ aside antiresorptives are generally well tolerated, with the exception of gastroesophageal irritation, although their administration require adequate calcium and vitamin D intake before and during therapy.
Kennel KA, Drake MT. Adverse effects of bisphosphonates: implications for osteoporosis management. Mayo Clin Proc. 2009;84(7):632-638.
As suggested we referred to the metastasis pattern of the explored cancer types to partially explain the increased MRONJ risk and added these considerations in the text:
The metastasis pattern of breast cancer varies with the subtype and has a predilection to metastasise to the bone the same seems to happen for prostate cancer when it stops responding to deprivation therapy.
These findings could help the oral surgeon in tailoring the most appropriate dental/oral prevention and follow-up to individual patients starting from cancer diagnosis and may be useful for stratifying patients taking antiresorptives agents when, for example, it is associated with osteoporosis induced by previously hormonal therapies.
Coleman RE, Rubens RD. The clinical course of bone metastases from breast cancer. Br J Cancer. 1987 Jan;55(1):61-6.
Berruti, A., Tucci, M., Mosca, A. et al. Predictive factors for skeletal complications in hormone-refractory prostate cancer patients with metastatic bone disease. Br J Cancer 93, 633–638 (2005).
The question about prostate cancer has been partially answered by the previous comment on the metastatic behavior of prostate cancer which combined with the long-term overall survival of these patients allow for prolonged expossure to bone antiresorptive agents.
The open question about antiresorptives discontinuation has been discussed as suggested adding the following considerations:
This is indeed a pertinent question as it was shown that the risk for osteonecrosis of the jaw increases with duration of bisphosponate therapy even if it is unclear whether there is sufficient evidence to support de-escalation as a standard of care.
Bamias A, Kastritis E, Bamia C, Moulopoulos LA, Melakopoulos I, Bozas G, Koutsoukou V, Gika D, Anagnostopoulos A, Papadimitriou C, Terpos E, Dimopoulos MA. Osteonecrosis of the jaw in cancer after treatment with bisphosphonates: incidence and risk factors. J Clin Oncol. 2005 Dec 1;23(34):8580-7.
M.F.K. Ibrahim, S. Mazzarello, R. Shorr, L. Vandermeer, C. Jacobs, J. Hilton, B. Hutton, M. Clemons, Should de-escalation of bone-targeting agents be standard of care for patients with bone metastases from breast cancer? A systematic review and meta-analysis. Annals of Oncology, Volume 26, Issue 11, 2015, Pages 2205-2213
In general, in the absence of contraindications and relevant side-effects patients should continue denosumab once every 4 weeks without any changes in the application regimen while patients with stable bone metastases on zoledronic acid, extending the administration frequency from once every 4 weeks to once every 12 weeks appears to be reasonable.
However, the ONJ rate was not reduced with this alternative regimen [Amadori et al. Lancet Oncol 2013;14:663-670].
Finally the question on the differences in the patient’s prognosis using different medication has been discussed
A recent systematic review and meta-analysis by Limones et al. found no statistically significant differences in the prognosis of MRONJ cases due to denosumab or zoledronic acid (P = 0.163). However, individual studies reported that MRONJ cases have a slightly higher tendency to resolve, ranging from 18 to 50% compared to ZA, which ranges from 8% to 43%. This might be related to the revers- ibility mechanism inherent to denosumab, which is not found in bisphosphonates in general.
Limones A, Sáez-Alcaide LM, Díaz-Parreño SA, Helm A, Bornstein MM, Molinero-Mourelle P. Medication-related osteonecrosis of the jaws (MRONJ) in cancer patients treated with denosumab VS. zoledronic acid: A systematic review and meta-analysis. Med Oral Patol Oral Cir Bucal. 2020 May 1;25(3):e326-e336. doi: 10.4317/medoral.23324. PMID: 32271321; PMCID: PMC7211372.
We hope this meet the reviewer's expectations.
Round 2
Reviewer 2 Report
Can be published.